# Transcriptome Differences in Normal Human Bronchial Epithelial Cells in Response to Influenza A pdmH1N1 or H7N9 Virus Infection

**DOI:** 10.3390/cells11050781

**Published:** 2022-02-23

**Authors:** Tzu-Hsuan Hsieh, Ya-Jhu Lin, Mei-Jen Hsioa, Hsin-Ju Wang, Lu-Ting Chen, Shu-Li Yang, Chung-Guei Huang

**Affiliations:** 1Department of Laboratory Medicine, Linkou Chang Gung Memorial Hospital, Taoyuan 33305, Taiwan; tcmmidi@cgmh.org.tw (T.-H.H.); niny@cgmh.org.tw (Y.-J.L.); caro10218@cgmh.org.tw (M.-J.H.); nessawan06@cgmh.org.tw (H.-J.W.); tina76205@gmail.com (L.-T.C.); shuli@cgmh.org.tw (S.-L.Y.); 2Department of Medical Biotechnology and Laboratory Science, College of Medicine, Chang Gung University, Taoyuan 33302, Taiwan; 3Research Center for Emerging Viral Infections, College of Medicine, Chang Gung University, Taoyuan 33302, Taiwan

**Keywords:** avian influenza A(H7N9) virus, pandemic influenza H1N1 (pdmH1N1), transcriptome

## Abstract

Avian influenza A (H7N9) virus infections frequently lead to acute respiratory distress syndrome and death in humans. The emergence of H7N9 virus infections is a serious public health threat. To identify virus–host interaction differences between the highly virulent H7N9 and pandemic influenza H1N1 (pdmH1N1), RNA sequencing was performed of normal human bronchial epithelial (NHBE) cells infected with either virus. The transcriptomic analysis of host cellular responses to viral infection enables the identification of potential cellular factors related to infection. Significantly different gene expression patterns were found between pdmH1N1- and H7N9-infected NHBE cells. In addition, the H7N9 virus infection induced strong immune responses, while cellular repair mechanisms were inhibited. The differential expression of specific factors observed between avian H7N9 and pdmH1N1 influenza virus strains can account for variations in disease pathogenicity. These findings provide a framework for future studies examining the molecular mechanisms underlying the pathogenicity of avian H7N9 virus.

## 1. Introduction

The influenza A virus is a negative-sense, single-stranded, and enveloped RNA virus that belongs to the *Orthomyxoviridae* family. Through gene rearrangement, influenza viruses swap gene segments, leading to the production of new viruses [1]. There have been several genetic recombination influenza pandemics in history, such as H2N2 in 1957 and H3N2 in 1968. The latest influenza virus pandemic was pandemic influenza H1N1 (pdmH1N1) in 2009. The pdmH1N1 virus is transmitted efficiently among humans and there were many patients with severe pdmH1N1 infection because of the widespread transmission [2]. The CDC estimated that approximately 151,700–575,400 people worldwide died from pdmH1N1 infection during the first year of the virus [3]. Avian-origin influenza A H7N9 infection in humans was initially reported in China in March 2013 [4,5,6]. A phylogenetic study demonstrated that the novel H7N9 virus originated from multiple reassortment events [7]. H7N9 transmission occurs from poultry to humans, with ducks and chickens likely acting as intermediate hosts. The ability of the avian influenza virus to break the species barrier and infect humans is determined by *HA* and *PB2*. Two key amino acid mutations, namely, G186V and Q226L, in HA increase the binding force of avian viruses to α2-6-linked sialidases, which are abundant on human respiratory epithelial cells, and the E627K mutation in PB2 facilitates replication of the virus in human cells [8,9]. Avian influenza A H7N9 virus infections frequently lead to acute respiratory distress syndrome, multiorgan disfunction, shock, and death in humans; there were 133 laboratory-confirmed H7N9 virus cases in March to July 2013, with 43 deaths reported, giving a case fatality of 32%. All fatal cases experienced pneumonia and acute respiratory distress syndrome [10]. To date, there have been 1564 confirmed cases of H7N9 infections with at least 612 deaths [11]. The emergence of H7N9 virus infections is a serious public health threat. When viruses attach to host cells, host defense mechanisms are triggered, and the innate immune system plays a major role. To identify virus–host interaction differences between the highly virulent H7N9 and pdmH1N1, RNA sequencing (RNA-seq) was performed on normal human bronchial epithelial (NHBE) cells infected with either virus. Importantly, the transcriptomic analysis of host cellular responses to viral infections enables the investigation of potential cellular factors related to viral infection [12,13]. Significantly different gene expression patterns were found between pdmH1N1- and H7N9-infected NHBE cells. We determined that H7N9 virus infection induced a strong immune response, while inhibiting cellular repair mechanisms. These findings provide a framework for future studies examining the molecular mechanisms underlying the pathogenicity of avian H7N9 virus.

## 2. Materials and Methods

### 2.1. NHBE Cells

NHBE cells from a 24-year-old female donor and BEGM™ Bronchial Epithelial Cell Growth Medium BulletKit™ culture medium (CC-3170) were obtained from Lonza (Walkersville, MD, USA). Cells at the third passage were grown in a confluent monolayer for subsequent infection experiments, as we described previously [14].

### 2.2. Virus Isolation and Preparation

A/Taiwan/4-CGMH2/2014 (H7N9) is genetically close to the A/Anhui/1/2013 strain, which was the fourth H7N9 infection reported on 25 April 2014 in Taiwan [15]. A/Taiwan/4-CGMH2/2014 (H7N9) and A/California/7/2009 (pdmH1N1)pdm09-like virus (CGMH-90126) were isolated from infected patients in Chang Gung Memorial Hospital in Taiwan. The viruses were propagated in 10–11-day-old embryonated eggs and incubated at 37 °C for 3 days, as we described previously [16]; subsequently, the allantoic fluid of inoculated chicken eggs was harvested and stored at −80 °C until use.

### 2.3. Virus Infection

NHBE cells were challenged by three multiplicity of infection (MOI) of H7N9, pdmH1N1, or mock control. After 1 h viral adsorption, the medium was removed and the cells were washed with phosphate-buffered saline before further incubation for 4, 8, 12, and 36 h in BEBM™ Bronchial Epithelial Basal Medium (CC-3171, Lonza), followed by the collection of culture medium supernatants and cell pellets.

### 2.4. RNA Extraction

The total RNA of the NHBE cell pellets collected from 12 and 36 h incubation with H7N9, pdmH1N1, or mock control was extracted by TRIzol™ Reagent (Thermo Fisher Scientific, Waltham, MA, USA) following the manufacturer’s instructions.

### 2.5. Influenza A Virus Quantification by Quantitative Real-Time Reverse Transcription Polymerase Chain Reaction

All procedures were conducted in biosafety level 2 facilities by personnel wearing biosafety level 3 personal protective equipment. Quantitative real-time reverse transcription polymerase chain reaction (qRT-PCR) was used for influenza A viral load quantitation, as we described previously [11]. Viral RNA was extracted from culture supernatants using a QIAamp Viral RNA Mini kit (Qiagen Inc., Valencia, CA, USA), followed by qRT-PCR. The primers and probe that targeted the conserved sequences on matrix gene (M) of influenza A used in this study were originally developed by the World Health Organization Collaborating Centre in Beijing, China (2013). The nucleotide sequences were as follows: forward primer, 5′-GACCRATCCTGTCACCTCTGAC-3′; reverse primer, 5′-AGGGCATTYTGGACAAAKCGTCTA-3′; and probe, 5′-FAM-TGCAGTCCTCGCTCACTGGGCACGBHQ1-3′. For quantification, plasmid DNAs at six different concentrations, ranging from 10–10^6^ copies/μL, were run in parallel with all samples.

### 2.6. Affymetrix Analysis

Human gene expression was examined using the GeneChip™ Human Transcriptome Array (HTA) 2.0 (Affymetrix, Santa Clara, CA, USA). HTA 2.0 maximizes the amount of unique and valuable information possible by minimizing the conserved sequence synthesized on the array. This high-resolution array design contains an unprecedented >6.0 million probes covering coding transcripts and non-coding transcripts. 70% of the probes on this array cover exons for coding transcripts, and the remaining 30% of probes on the array cover exon–exon splice junctions and non-coding transcripts. The unparalleled coverage of this array provides the deepest insight into all coding and non-coding transcripts available. RNA quality control, sample labeling, GeneChip hybridization, and data acquisition were performed at the Genomic Medicine Research Core Laboratory at Chang Gung Memorial Hospital in Linkou. The total RNA quality was checked using an Agilent Technologies 2100 Bioanalyzer (Santa Clara, CA, USA). The RNA was then amplified and labeled using a GeneChip^®^ WT Sense Target Labeling and Control Reagents kit (Affymetrix, Santa Clara, CA, USA). cDNA was synthesized, labeled, and hybridized to the GeneChip array according to the manufacturer’s protocol. Hybridization controls were used to assess hybridization quality. The arrays were washed and stained using a GeneChip Fluidics Station 450 (Affymetrix) and then scanned using a GeneChip Scanner 3000 7G (Affymetrix). GeneSpring GX 11 (Agilent) was used for the normalization, filtering, and statistical data analysis of microarray data. The linear data were first summarized using the Exon Robust Multichip Average (RMA) summarization algorithm on the CORE probe sets and baseline transformation to the median of all samples for the three major tasks, background correction, normalization, and probe summarization. We also used Exon RMA for GC-based background correction.

### 2.7. Bio-Plex Cytokine Analysis

Culture supernatants were collected at 4, 8, and 12 h and assayed for cytokines using a Bio-Plex Cytokine panel (Bio-Rad Laboratories, Hercules, CA, USA) and analyzed using a Bio-Plex Luminex 200 (Bio-Rad Laboratories). The assay was performed according to the manufacturer’s instructions to measure the concentrations of the following 27 target cytokines: IL-1β; IL-1 receptor antagonist; IL-2; IL-4; IL-5; IL-6; IL-7; IL-8; IL-9; IL-10; IL-12p70; IL-13; IL-15; IL-17A; basic fibroblast growth factor; eotaxin; granulocyte colony-stimulating factor (G-CSF); granulocyte macrophage CSF (GM-CSF); IFN-γ; IFN-γ-induced protein 10; monocyte chemotactic protein 1 (MCP-1); macrophage inflammatory protein 1α (MIP-1α); MIP-1β; platelet-derived growth factor BB; regulated on activation, normal T cell expressed and secreted (RANTES); TNF-α; and vascular endothelial growth factor (VEGF). The samples were incubated with antibody-coupled beads for 60 min followed by incubation with a detection antibody for 30 min. Next, conjugates were incubated with streptavidin for 10 min, washed using a Bio-Plex Pro II Wash Station (Bio-Rad Laboratories), resuspended, and vortexed before fluorescence measurement using a Bio-Plex^®^ 200 system (Bio-Rad Laboratories). The obtained data were analyzed and standard curves (log (x) − linear(y)) were generated using Bio-Plex Manager v6.0. The cytokine concentration levels were measured in triplicate and compared against standard curves generated by Bio-Plex Manager v6.0. Correction and quantile normalization were then performed using a median polish probe summarization. In addition to the quality control of the RNA samples and hybridization, principal component analysis was performed to check the data quality. Only data of those samples found to be satisfactory in all quality control tests were included for further analysis. In the process of data filtering, probe sets with an intensity value of the lowest 20th percentile of all intensity values were removed. The resulting working transcript list of filtered entities was then used for statistical analysis. Analysis of variance (ANOVA) was performed to identify those genes that were significantly expressed (*p* < 0.05) in response to viral infection. To reduce the overall number of false positives, Benjamini and Hochberg multiple testing correction was employed. Significantly differentially expressed genes (DEGs) with a fold change more than 1.5 in response to pdmH1N1 and seasonal pdmH1N1 infection compared with mock were then merged into a gene list for further gene ontology (GO) and pathway analysis. GO and pathway over-representation analysis, as well as further analysis of protein–protein interactions and transcription factor regulation, were performed using the Innate DB platform. Over-representation analysis was performed using a hypergeometric algorithm, and over-represented GO terms or pathways with *p*-values ≤ 0.05 were retained, provided that at least two of the uploaded genes mapped to the entity in question.

## 3. Results

### 3.1. Viral RNA Quantities Comparison of Influenza A pdmH1N1 and H7N9 in NHBE Cells

We compared the viral RNA quantities of influenza A pdmH1N1 and H7N9 in NHBE cells (Figure 1). NHBE cultures were infected with each virus at an MOI of 3, and culture supernatants at 4, 8, 12, and 36 h post-infection (hpi) were collected and viral RNA copies evaluated using qRT-PCR. We found a higher viral load of pdmH1N1 than H7N9 at 4 hpi, with a *p*-value of 0.0293, but a higher viral load of H7N9 than pdmH1N1 at 8, 12, and 36 hpi, with *p*-values of 0.0177, 0.0075, and 0.0104, respectively. This indicates that the growth of H7N9 can reach higher titers than pdmH1N1.

### 3.2. Global Overview of RNA-Seq Data of pdmH1N1- and H7N9-Infected NHBE Cells

The NHBE cells were harvested at 12 and 36 hpi, and RNA-seq was performed. We identified a total of 67,528 differentially expressed genes (DEGs) from a comparison among mock and virus-infected groups that were either twofold upregulated or downregulated. The DEGs were identified using a false discovery rate *q*-value threshold of less than 0.05 (Table 1). In our analysis, we found that the number of DEGs in the pdmH1N1-infected group decreased during infection. At 12 hpi, there were 3448 DEGs in pdmH1N1-infected cells, representing about 5% of all genes. Of these DEGs, the proportion of upregulated and downregulated DEGs was 71% and 29% (2451/3448 and 997/3448), respectively. At 36 hpi, there were 1213 DEGs in pdmH1N1-infected cells, representing about 2% of the total genes; however, the proportion of upregulated DEGs increased to 92% (1122/1213), with a corresponding decrease to 8% (91/1213) of the downregulated DEGs. In comparison, there were more DEGs in the H7N9-infected cells, and the number of DEGs increased during infection. At 12 hpi, we identified 10,353 DEGs in the H7N9-infected cells, comprising approximately 15% of all genes. Of these DEGs, we found that 81% (8379/10,353) were upregulated and 19% (1974/10,353) were downregulated. At 36 hpi, there were 12,669 DEGs in the H7N9-infected cells, representing about 19% of the total genes, and the proportion of upregulated and downregulated DEGs was 78% (9940/12,699) and 22% (2729/12,699), respectively (Figure 2A). Using a Venn diagram to illustrate the overlapping DEG profiles for pdmH1N1 and H7N9 (Figure 2B), we found 319 and 5215 overlapping DEGs between 12 and 36 hpi for the pdmH1N1 and H7N9 groups, respectively. In addition, there were 1665 and 829 overlapping DEGs between the two groups at 12 and 36 hpi, respectively. Volcano plots show that the number of DEGs in the H7N9-infected group was significantly higher than the pdmH1N1-infected group (Figure 2C). Based on statistical enrichment analysis, we identified and listed the top 10-fold-change upregulated and downregulated DEGs by the absolute value of the log base 2 scale obtained from both groups (Table 2). Gene expression analysis revealed that the interferon-related genes, *IFI6*, *IFI44L*, *IFIT1*, *IFIT3*, *IFI44*, and *IFIT2*, were upregulated in NHBE cells infected with pdmH1N1 at 12 hpi, but the effect on these interferon-related genes declined over time. There were more RNA synthesis-related genes upregulated at 36 hpi, especially *VTRNA1-3*, which had a fold change of 1663. The most upregulated genes in NHBE cells infected with H7N9 were similar at 12 hpi, and the top three upregulated genes were *RP1-12G14.6*, *RNA5SP115*, and *POLG2*. We also determined that the top downregulated genes were similar in NHBE cells infected with pdmH1N1 or H7N9. Those genes showing the greatest expression decrease were the cellular repair pathway-related genes, *KRT4*, *RPTN*, and *PPL*. In fact, the expression of these genes was significantly inhibited in H7N9-infected NHBE cells, with *KRT4* showing a fold change of −640.

### 3.3. Canonical Signaling Pathways Analysis Based on DEGs

The Ingenuity Pathway Analysis (IPA) tool was used to generate a list of the most significant canonical pathways and the highest activated networks with their respective IPA scores. In pdmH1N1-infected NHBE cells, the IPA scores of both upregulated and downregulated pathways were at zero-fold at 36 hpi, reflecting a minimal change in the pdmH1N1-infected group. In contrast, we identified pathways with fold changes that continued to change over time in the H7N9-infected group. For example, both the PTEN signaling and HIPPO signaling pathways increased over time, whereas the canonical pathways of IL-17A signaling in airway cells, signaling by Rho family GTPases, p70S6K signaling, thrombin signaling, role of NFAT in cardiac hypertrophy, ErbB2-ErbB3 signaling, and leukocyte extravasation signaling continued to decrease.

The results of our canonical signaling pathway analysis showed that the interferon signaling pathway and apoptosis pathway were highly activated in H7N9-infected NHBE cells (Figure 3A). The interferon signaling pathway upregulation in the pdmH1N1-infected group at 12 hpi was significantly greater than that of the H7N9-infected group, but it had almost recovered at 36 hpi. The reduced activation of the apoptosis pathway in the pdmH1N1-infected group may explain the mild symptoms in humans. Further, we also used IPA to investigate related genes in the interferon signaling pathway (Figure 4) and apoptosis pathway (Figure 5).

### 3.4. Differential Cytokine Expression in NHBE Cells Challenged by pdmH1N1 or H7N9

To further understand the cell reactions triggered by pdmH1N1 or H7N9 infection, we also used a Bio-Plex Cytokine panel to investigate the concentrations of 27 target cytokines, each of which was classified to one of the following five groups: chemokines (IL-8, IP-10, RANTES, MIP-1β, eotaxin, MCP-1, and MIP-1α), growth factors (PDGF-BB, FGF basic, GM-CSF, VEGF, and G-CSF), proinflammatory cytokines (IL-1β, IL-6, and TNF-α), T-helper cytokines (IL-12, IL-2, IL-5, IL-9, IL-17, IFN-γ, IL-10, IL-4, and Il-13), and others (IL-1Ra, Il-7, and IL-15) (Table 3). We found that H7N9 infection induced the expression of chemokines IL-8, IP-10, and RANTES in NHBE cells and reduced the expression of MIP-1β, eotaxin, MCP-1, and MIP-1α. No expression change of MIP-1β, eotaxin, and MCP-1 was found in the pdmH1N1-infected group. Further, we found the opposite in the expression of growth factors GM-CSF and G-CSF between the two groups. Similarly, the expression of the proinflammatory cytokines IL-1β, IL-6, and TNF-α was induced by H7N9 infection, but decreased in response to pdmH1N1 infection. The expression of all T-helper cytokines in the H7N9-infected group decreased, but a 2–91 fold change was found in the expression of IFN-γ, IL-10, IL-4, and IL-13 in the pdmH1N1-infected group. We also found that the expression of proinflammatory and T-helper cytokines in H7N9- and pdmH1N1-infected cells was significantly different (Figure 6).

## 4. Discussion

Our examination of the viral RNA quantities of influenza A pdmH1N1 and H7N9 in NHBE cells showed that H7N9 replicates more effectively in humans than pdmH1N1. Similar results were reported in human A549 cells, in which the H7N9 virus replicates faster and to higher titers than pdmH1N1 strains [17]. A comparison between patients infected with H7N9 and pdmH1N1 complicated by acute respiratory distress syndrome shows that the H7N9-infected group had a longer duration of viral shedding from the onset of illness and from the initiation of antiviral therapy to a negative viral test result than the pdmH1N1-infected group [18]. Further, H7N9-infected patients had a longer duration of hospitalization than pdmH1N1-infected patients, and the median time from onset to death was 18 days vs. 15 days for H7N9 and pdmH1N1, respectively [19].

According to RNA-seq analysis in this study, the number of genes (DEGs) affected by H7N9 was extremely large. From 12 to 36 hpi, the number of affected genes continued to increase, whereas in contrast, fewer genes were affected by pdmH1N1 infection and the number of DEGs decreased with time. The highest expressed DEGs in response to H7N9 infection were *RP1-12G14.6*, *RNA5Sp115*, and *POLG2* at 12 and 36 hpi, whereas the lowest expressing genes were cell repair-related genes, such as *KRT4*, *RPTN*, and *PPL*. Our findings indicate that H7N9 has a greater impact on cells, and longer time course and the ability of cells to repair may be limited in H7N9-infected NHBE cells.

Following influenza A viral infection, the host cell triggers a complex regulatory system of innate and adaptive immune responses to defend against the virus. One of the many responses to the viral invasion is the induction of interferons, which function by stimulating cytotoxic T cells, and by inducing a number of intracellular genes that directly prevent virus replication or facilitate apoptosis [20,21,22,23]. Based on pathway-level IPA analysis, we found that IFN-related pathways were activated in NHBE cells infected with either H7N9 or pdmH1N1. At 12 hpi, the degree of activation of IFN-related pathways was greater in the pdmH1N1-infected group. We also found increased expression of IFN-γ in our cytokine analysis in the pdmH1N1-infected group at 8 and 12 hpi compared with the H7N9-infected group. Interestingly, the expression of RNA synthesis-related genes was more prominent at 36 hpi than IFN-related pathways in NHBE cells infected with pdmH1N1, indicating that pdmH1N1 infection may be efficiently controlled in NHBE cells by appropriately elevated IFN-related responses in the early infection stage.

According to the pathway-level IPA analysis, we found that the apoptosis pathway was highly activated in H7N9-infected NHBE cells. Lee et al. [24] showed that H7N9-infected monocytes died rapidly via apoptosis, but pdmH1N1-infected monocytes did not. The IL-17A signaling in airway cells was downregulated in H7N9-infected NHBE cells in IPA analysis, and IL-17 was negatively expressed in our cytokine analysis. The result is consistent with other studies: Bao et al. [25] claimed that the IL-17A protein and mRNA levels were decreased in patients with H7N9 infection and a restored Th17 and Tc17 cell frequency might serve as a biomarker for disease progression in patients infected with this virus.

We also discovered that the HIPPO pathway was highly activated in H7N9-infected NHBE cells, and the degree of activation continuously increased at 12–36 hpi. At 36 hpi, HIPPO was the most activated pathway in H7N9-infected NHBE cells, but there was no expression of HIPPO signaling in cells from the pdmH1N1-infected group. The HIPPO pathway plays critical roles in the regulation of innate immunity. The core components of the canonical HIPPO pathway in mammals consist of mammalian Ste20-like kinases 1/2 (MST1/2) and their downstream effectors LATS and NDR, which limit proinflammatory cytokine (IL-6 and TNFα) production by inhibiting IRAK1/NF-κB and Mekk2. During viral infection, HIPPO signaling blocks the negative regulatory effect of virus-activated kinase IKK phosphorylation of YAP on antiviral immunity. The HIPPO pathway plays important roles in innate immunity against pathogens and protects the host from inflammatory injury during infection [26]. In the early stage of H7N9 infection, the HIPPO pathway activates in avian cells, but not in human cells. The pathway allows the H7N9 virus to remain in its low pathogenicity form in the avian host, resulting in a non-diseased state during an H7N9 epidemic [27]. Interestingly, we found activation of the HIPPO pathway in NHBE cells in the middle and late stages of H7N9 infection, but not in pdmH1N1 infection. The concentration of cytokines in NHBE cells in response to infection in our study indicated that the expression of the proinflammatory cytokines IL-1β, IL-6, and TNF-α was induced by H7N9 infection, but reduced by pdmH1N1 in the early stage of infection. Similarly, Lee et al. [24,28] also showed that H7N9-infected peripheral blood mononuclear cells had significantly higher mRNA levels of proinflammatory cytokines and type I interferons at 6 hpi.

The expression of proinflammatory and T-helper cytokines in H7N9- and pdmH1N1-infected cells was significantly different. As a result, in the early stage of infection, T-helper cytokines were highly triggered in the pdmH1N1-infected group to defend against the infection, and there was a negative proinflammatory reaction to protect the host cell from inflammation injury. Although H7N9 infection induced the stable and continuous activation of IFN-related pathways, there was a negative expression of T-helper cytokines in the early stage of infection. On the contrary, the proinflammatory factors were highly activated; we hypothesize that there is a lack of control of H7N9 infection in human cells in the early stage of infection, and the host cell must trigger a more efficient immune regulation against the infection, thus activating the HIPPO pathway. A report about the clinical outcome in patients with the H7N9 infection indicated that high chemokine and cytokine levels were observed [29,30]. Our findings indicate that H7N9 causes a strong proinflammatory cytokine and chemokine reaction at the early stage of infection, leading to a cytokine storm and causing greater symptom severity in human hosts.

## 5. Conclusions

In this study, we determined that there were significantly different gene expression patterns between pdmH1N1- and H7N9-infected NHBE cells. H7N9 virus infection induced a strong immune response, while inhibiting cellular repair mechanisms. The differential expression of specific factors observed between avian H7N9 and pdmH1N1 influenza virus strains can explain variations in disease pathogenicity. These findings provide a framework for future studies examining the molecular mechanisms underlying the pathogenicity of avian H7N9 virus. In addition, the results of our study provide valuable information regarding the virus–host interaction between H7N9 and NHBE cells, which improves our understanding of the pathogenic mechanisms that lead to severe complications. Collectively, our data provide a new insight into the underlying mechanisms of the differential pathogenicity of avian influenza viruses.

## Figures and Tables

**Figure 1 cells-11-00781-f001:**
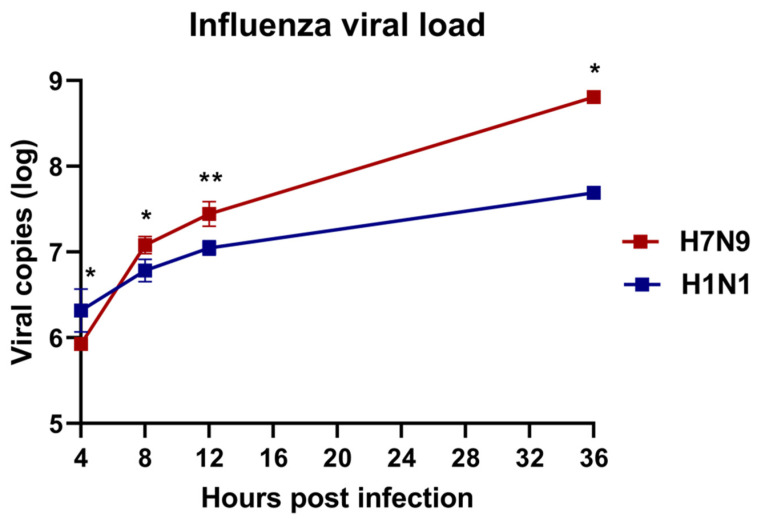
Normal human bronchial epithelial (NHBE) cell cultures were infected with pandemic influenza H1N1 (pdmH1N1) and H7N9 at a multiplicity of infection of three and performed in triplicate. Culture supernatants at 4, 8, 12, and 36 h post-infection (hpi) were collected and the viral load was evaluated using quantitative real-time reverse transcription polymerase chain reaction (* = *p*-value 0.01–0.05, significant; ** = *p*-value 0.001–0.01, very significant).

**Figure 2 cells-11-00781-f002:**
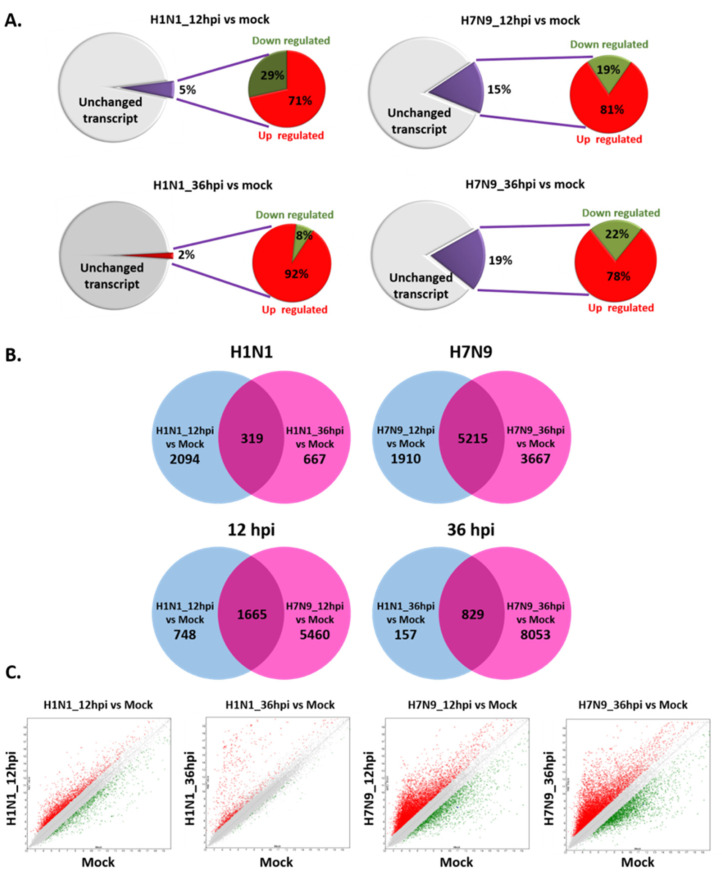
A global overview of RNA-seq data of NHBE cells infected with mock, pdmH1N1 virus, or the H7N9 virus. The NHBE cells were harvested at 12 and 36 hpi, and RNA-seq was performed. (**A**) The number of upregulated and downregulated DEGs with more than twofold change identified from comparisons between the mock and virus-infected groups. DEGs were identified using an FDR *q*-value threshold of less than 0.05. (**B**) Venn diagrams of overlapping DEG profiles for pdmH1N1- and H7N9-infected groups. Displayed DEGs have a twofold change or more with a *p*-value of less than or equal to 0.05. Differential expression of upregulated and downregulated mRNAs in pdmH1N1- and H7N9-infected NHBE cells are depicted in two overlapping circles at 12 and 36 hpi. Values indicate the mRNA counts in the indicated areas. (**C**) Volcano plots showing DEGs for pdmH1N1- and H7N9-infected groups. The *x*-axis represents the log base 2 values of the fold change observed for each mRNA transcript, and the *y*-axis represents the log base 10 values of *p*-values of significance tests between replicates for each transcript. Data for genes not classified as differentially expressed are plotted in black.

**Figure 3 cells-11-00781-f003:**
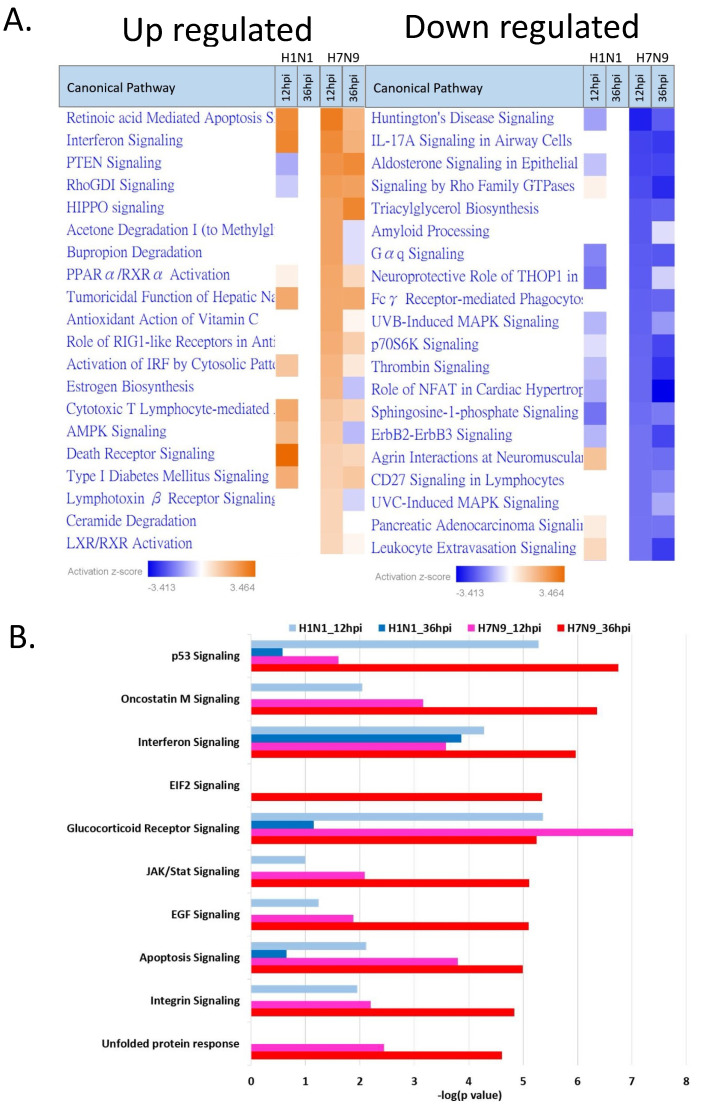
Top canonical signaling pathways activated in NHBE cells by pdmH1N1 and H7N9 infection. (**A**) The Ingenuity Pathway Analysis (IPA) tool was used to identify and list the most significant canonical pathways and highest activated networks with their respective IPA scores. (**B**) The top 10 upregulated canonical signaling pathways activated by H7N9 at 36 hpi.

**Figure 4 cells-11-00781-f004:**
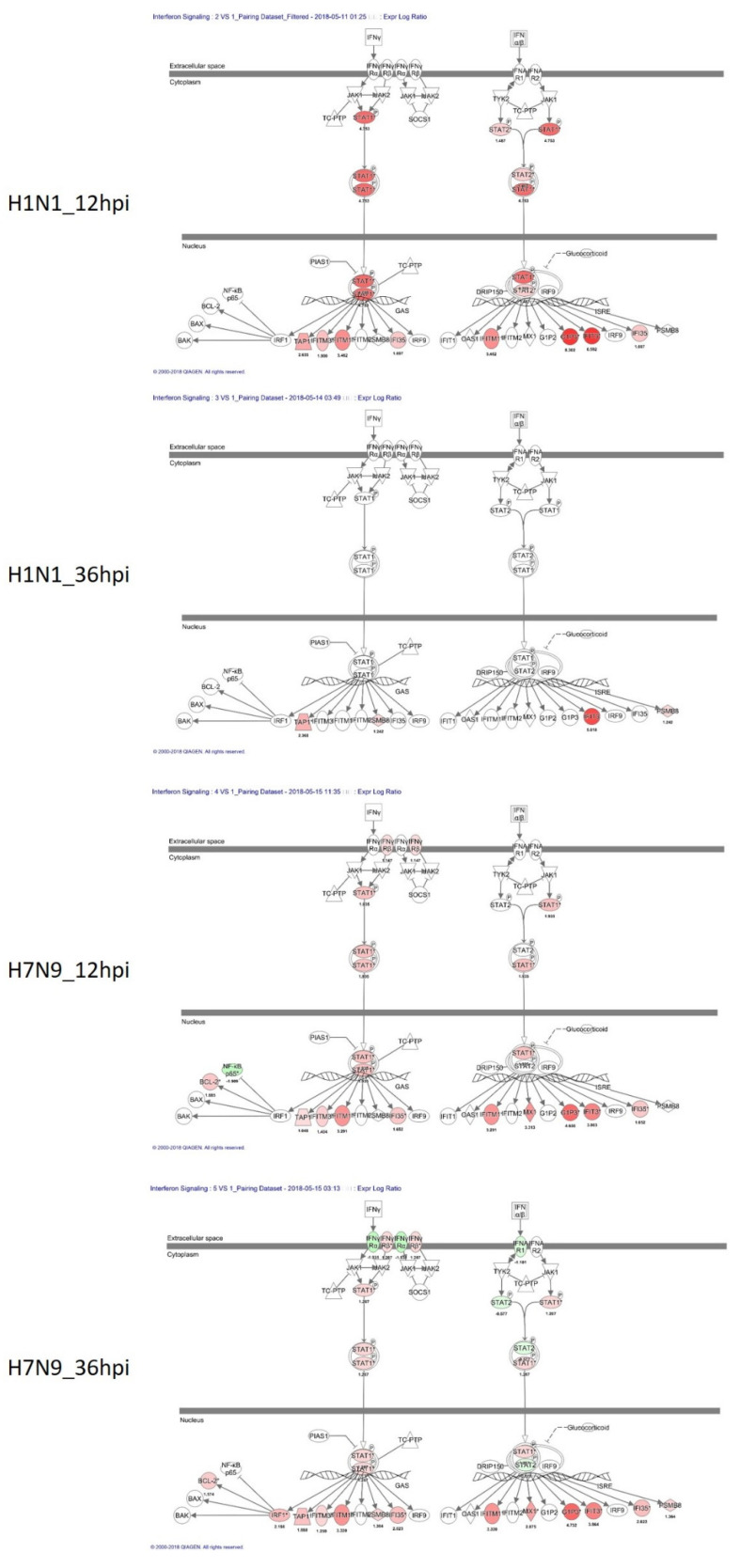
Genes associated with the interferon signaling canonical pathway. IPA identified those pathways that were differentially expressed between pdmH1N1- and H7N9-infected NHBE cells. DEGs associated with the interferon signaling canonical pathway are shown in color. The color intensity indicates the degree of upregulation (red) or downregulation (green) relative to mock-infected NHBE cells. Solid lines represent direct interactions and dashed lines show indirect interactions.

**Figure 5 cells-11-00781-f005:**
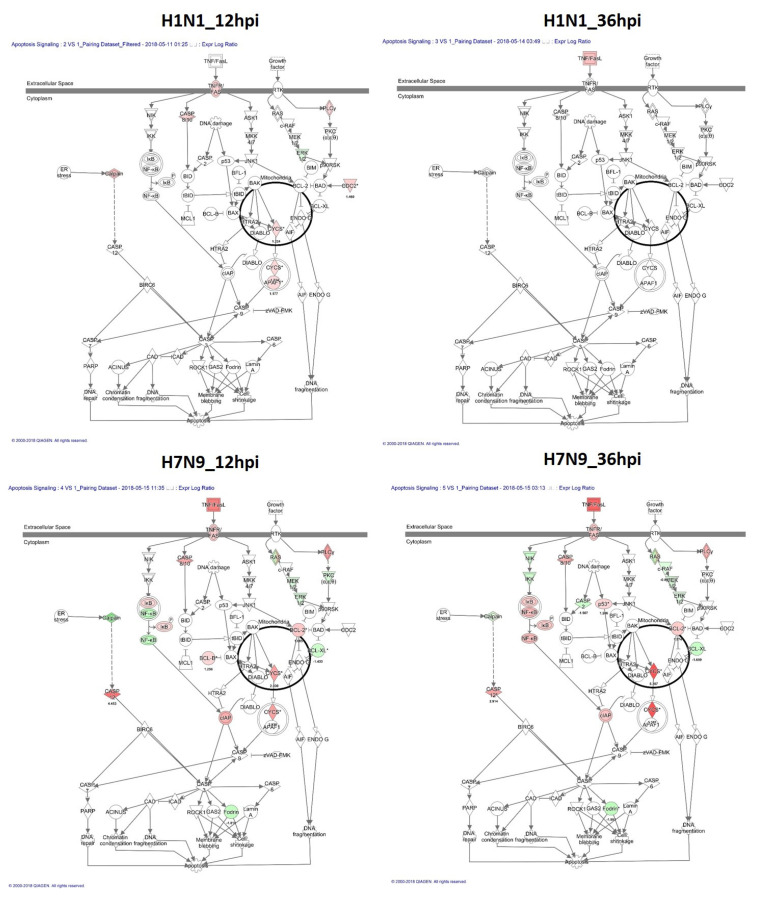
Genes associated with the apoptosis signaling canonical pathway (bold circle). IPA pathway analysis identified pathways that were differentially expressed between pdmH1N1- and H7N9-infected NHBE cells. DEGs associated with the apoptosis signaling canonical pathway appear in color. The color intensity indicates the degree of upregulation (red) or downregulation (green) relative to mock-infected NHBE cells. Solid lines represent direct interactions and dashed lines show indirect interactions.

**Figure 6 cells-11-00781-f006:**
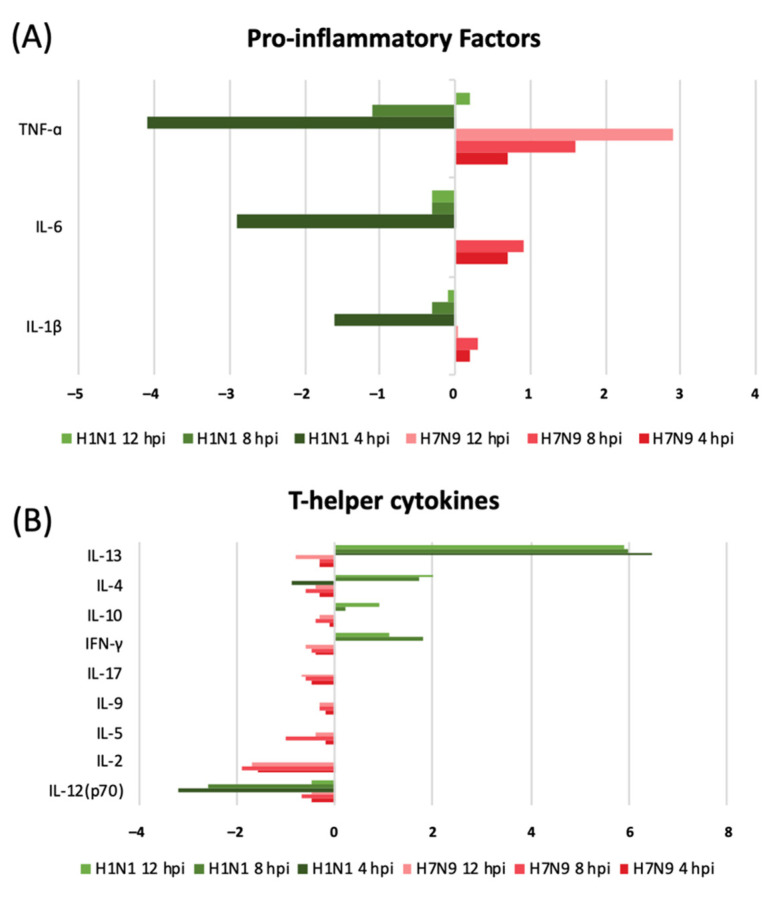
The differential expression of proinflammatory factors and T-helper cytokines in NHBE cells challenged by pdmH1N1 or H7N9 infection. Culture supernatants collected from NHBE cells infected with H7N9 and pdmH1N1 at 4, 8, and 12 hpi were assayed for cytokines. (**A**) The differential expression of the proinflammatory factors TNF-α, IL-6, and IL-1β. (**B**) The differential expression of the T-helper cytokines IL-13, IL-4, IL-10, IFN-γ, IL-17, IL-9, IL-5, IL-2, and IL-12p70. Values shown indicate the log base 2 ratio.

**Table 1 cells-11-00781-t001:** Overview of RNA sequencing (RNA-seq) data.

Total Gene No. 67,528	Upregulation		Downregulation		Significant Gene No.	% in Total Gene
	Coding	Non-Coding	Coding	Non-Coding		
H1N1_12hpi VS Mock	1678	773	735	262	3448	5%
H1N1_36hpi VS Mock	908	214	78	13	1213	2%
H7N9_12hpi VS Mock	5559	2820	1566	408	10,353	15%
H7N9_36hpi VS Mock	6727	3213	2155	574	12,669	19%

The number of upregulated and downregulated DEGs with more than twofold change identified from comparisons between the mock and virus-infected groups. DEGs were identified using a false discovery rate (FDR) *q*-value threshold of less than 0.05.

**Table 2 cells-11-00781-t002:** Top 10 upregulated and downregulated DEGs at 12 and 36 hpi.

Upregulation							Downregulation						
hpi	H1N1	Fold Change	*p*-Value	H7N9	Fold Change	*p*-Value	hpi	H1N1	Fold Change	*p*-Value	H7N9	Fold Change	*p*-Value
12 hpi	IFI6	315.62	0.00854	RP1-12G14.6	944.41	0.000383	12 hpi	CRNN	−293.19	0.003856	ATP12A	−2305.39	0.000245
	IFI44L	191.33	0.0036	RNA5SP115	435.61	0.00153		LCE3D	−273.6	0.017274	CNFN	−1263.62	0.010624
	IFIT1	138.87	0.006031	POLG2	372.61	0.000438		CNFN	−194.27	0.024209	KLK7	−751.41	0.004461
	MX1	137.1	0.003247	AMPD1	195.68	0.000091		FLG	−149.21	0.012985	CEACAM5	−749.71	0.001844
	IFIT3	96.49	0.007147	LINC00641	167.28	0.006579		NCCRP1	−140.35	0.001799	SBSN	−719.43	0.009373
	IFI44	88.53	0.001676	MSH4	164.85	0.000503		KLK7	−118.29	0.008553	NCCRP1	−642.08	0.001569
	RSAD2	86.66	0.015444	MIR215	158.55	0.00367		SBSN	−117.62	0.019107	KRT4	−598.76	0.000358
	DDX60L	81.55	0.001921	RNY5P3	152.08	0.004924		LCE3E	−113.71	0.007499	TMPRSS11E	−574.16	0.007775
	IFIT2	76.81	0.015337	CKAP2L	123.31	0.004605		ATP12A	−110.6	0.003672	IVL	−497.27	0.000799
	CXCL11	58.57	0.01463	LDHC	107.28	0.000013		WFDC12	−102.89	0.004885	CRNN	−435.42	0.003444
36 hpi	VTRNA1-3	1663.32	0.017978	RNA5SP115	1176.93	0.000935	36 hpi	CRNN	−21.31	0.03233	ATP12A	−1273.8	0.000371
	IFI6	738.27	0.062186	RP1-12G14.6	1164.64	0.000848		WFDC12	−10.05	0.040629	KRT4	−640.49	0.000352
	RNA5SP402	603.17	0.005802	POLG2	953.23	0.00052		IL36A	−7.45	0.027829	CEACAM5	−533.67	0.002502
	RNA5SP496	417.73	0.014153	RP11-505P4.6	647.16	0.000201		SPINK5	−5.73	0.019001	SBSN	−489.11	0.01055
	MIR4521	314.04	0.01979	MIR4659A	508.36	0.001933		GCNT3	−4.97	0.042463	NCCRP1	−486.72	0.001416
	RSAD2	267.72	0.045687	HIST2H4B	362.02	0.001517		SCNN1B	−4.77	0.022182	KLK7	−410.65	0.005305
	TRNAI6	255.48	0.000532	VTRNA1-3	326.31	0.024592		IVL	−4.63	0.046796	RPTN	−404.42	0.073667
	RNA5SP318	248.85	0.011112	HCP5	324.61	0.001104		KCNH5	−4.59	0.032183	CNFN	−401.92	0.015334
	TRNAI2	243.8	0.002995	RP1-40E16.11	260.6	0.000742		CLCA4	−4.33	0.031389	TMPRSS11E	−380.78	0.008919
	RSAD2	198.39	0.016356	RGS2	257.86	0.00157		SH3BGRL2	−3.96	0.028623	PPL	−376.61	0.010291

**Table 3 cells-11-00781-t003:** The differential expression of cytokines in NHBE cells challenged by pdmH1N1 or H7N9 infection.

Category	Name	H7N9			pdmH1N1		
		4 hpi	8 hpi	12 hpi	4 hpi	8 hpi	12 hpi
Chemokines	IL-8	0.1	0.9	1.0	−2.7	−0.8	−0.1
	IP-10	0.8	3.7	4.3	−1.3	3.3	4.2
	RANTES	−0.2	0.3	0.9	0.0	0.0	4.5
	MIP-1β	−0.6	−0.9	−0.2	0.0	0.0	0.0
	Eotaxin	−0.4	−0.8	−0.5	0.0	0.0	0.0
	MCP-1(MCAF)	−0.5	−1.1	−0.8	0.0	0.0	0.0
	MIP-1α	−0.1	−0.1	−0.3	−0.5	−0.5	−0.1
Growth Factors	PDGF-BB	−0.6	−1.1	−1.6	0.0	0.0	0.0
	FGF basic	−0.1	−0.2	−0.4	1.3	−2.9	−0.1
	GM-CSF	−0.1	−0.4	−0.3	−0.2	0.01	0.05
	VEGF	−0.1	−0.1	0.2	−1.9	−1.4	−0.6
	G-CSF	0.3	0.5	0.2	−9.5	−2.1	−1.0
Pro inflammatoryFactors	IL-1β	0.2	0.3	0.02	−1.6	−0.3	−0.1
	IL-6	0.7	0.9	0.0	−2.9	−0.3	−0.3
	TNF-α	0.7	1.6	2.9	−4.1	−1.1	0.2
T-helper cytokines	IL-12(p70)	−0.5	−0.7	−0.5	−3.2	−2.6	−0.5
	IL-2	−1.6	−1.9	−1.7	0.0	0.0	0.0
	IL-5	−0.2	−1.0	−0.4	0.0	0.0	0.0
	IL-9	−0.2	−0.3	−0.3	0.0	0.0	0.0
	IL-17	−0.5	−0.6	−0.7	0.0	0.0	0.0
	IFN-γ	−0.4	−0.5	−0.6	0.0	1.8	1.1
	IL-10	−0.1	−0.4	−0.3	0.0	0.2	0.9
	IL-4	−0.3	−0.6	−0.4	−0.9	1.7	2.0
	IL-13	−0.3	−0.3	−0.8	6.5	6.0	5.9
Others	IL-1Ra	0.0	−0.3	−0.3	0.1	0.1	0.1
	IL-7	−0.5	−1.1	−0.9	0.0	0.0	0.0
	IL-15	0.0	0.0	0.0	0.0	0.0	0.0

We classified 27 substances into the following five categories: chemokines, growth factors, proinflammatory factors, T-helper cytokines, and others. In the chemokine analysis of H7N9-infected cells, IL-8, IP-10, and RANTES showed an upward trend, whereas MIP-1β, eotaxin, MCP-1 (also known as MCAF), and MIP-1α showed a decline in the H7N9-infected group. There was no change in MIP-1β, eotaxin, and MCP-1 in the pdmH1N1-infected group, indicating different responses than in H7N9 infection. Growth factor analysis showed the opposite in GM-CSF and G-CSF between the two groups. Additionally, IL-1β, IL-6, and TNF-α showed an increased response in the H7N9-infected group, all of which were inhibited in the pdmH1N1-infected group. The expression of all T-helper cytokines decreased in the H7N9-infected group, but four factors increased in the pdmH1N1-infected group. The values shown indicate the log base 2 ratio. The color indicates the changes of expression: increased (red), decreased (green), and consistent (yellow).

## Data Availability

The data presented in this study are openly available in FigShare at doi: 10.6084/m9.figshare.19208673.

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
