# Peer review of "Transcriptome Differences in Normal Human Bronchial Epithelial Cells in Response to Influenza A pdmH1N1 or H7N9 Virus Infection"

_cells, 2022, doi:10.3390/cells11050781_

Round 1

Reviewer 1 Report

Tzu-Hsuan Hsieh and team has revealed the transcriptomic analysis of NHBE cells from a human donor while H7N9 and pdmH1N1 infections. Their findings sound provide a framework for future studies in H7N9 pathogenicity. 

Major:

  1. According to the qPCR results, the study team claimed that H7N9 is grew faster than pdmH1N1. However, the growth kinetics showed in Fig 1 seems not support the conclusion. Please provide a kinetics slope and analysis. In addition, the inoculate virus titters are same for both viruses?
  2. Interesting, there were no common up and down regulated gene found either 12 or 36 hpi for both viruses, at least in top 10, but they were some repeating gene found in the same time point by same virus (e.g. FLG, TMPRSS11E, HIST2H4B), were they not the same gene? It will be good to provide a full list of hits as supplementary.
  3. The study team has postulated a canonical signalling pathway based on DEGs and highlighted their degree level of changes. After which, there is lack of evidence to support their postulation is true.
  4. In discussion, the lines 319-323 was not support the statements from lines 309-318.
  5. The discussion of cytokines findings with canonical pathway is weak or no linked.

Minor:

  1. Please improve the introduction with using a number or range instead of “many” or “frequently”. For example, line 35 and 44.
  2. Since a subject donor has been indicated in section 2.1, please include the IRB approval.
  3. Should not a new paragraph at line 274.

Author Response

Response to Reviewer 1 Comments

Major:

Point 1: According to the qPCR results, the study team claimed that H7N9 is grew faster than pdmH1N1. However, the growth kinetics showed in Fig 1 seems not support the conclusion. Please provide a kinetics slope and analysis. In addition, the inoculate virus titters are same for both viruses?

Response 1: The inoculate virus titers are same for both viruses, we infected both viruses at a multiplicity of infection of three. We modified the statements about result 3.1 and Figure 1, we removed the “growth kinetics” in the text.

“3.1. Viral RNA quantities comparison of Influenza A pdmH1N1 and H7N9 in NHBE Cells

We compared the viral RNA quantities of influenza A pdmH1N1 and H7N9 in NHBE cells (Figure 1). NHBE cultures were infected with each virus at an MOI of 3, and culture supernatants at 4, 8, 12 and 36 h postinfection (hpi) were collected and viral RNA copies was evaluated using qRT-PCR. We found a higher viral load of pdmH1N1 than H7N9 at 4 hpi, with a p-value 0.0293. But a higher viral load of H7N9 than pdmH1N1 at 8, 12 and 36 hpi, with p-value 0.0177, 0.0075 and 0.0104 respectively. Indicating that the growth of H7N9 can reach higher titers than pdmH1N1.”

Point 2: Interesting, there were no common up and down regulated gene found either 12 or 36 hpi for both viruses, at least in top 10, but they were some repeating gene found in the same time point by same virus (e.g. FLG, TMPRSS11E, HIST2H4B), were they not the same gene? It will be good to provide a full list of hits as supplementary.

Response 2: The repeating genes FLG, TMPRSS11E and HIST2H4B in table 2 were mistakes caused by data arrangement. We updated the correct contents about the top 10 up and down DEGs in table 2.

Point 3: The study team has postulated a canonical signaling pathway based on DEGs and highlighted their degree level of changes. After which, there is lack of evidence to support their postulation is true.

Response 3: In this study, we analysis the gene expression level by GeneChip™ Human Transcriptome Array, and analysis the cytokine level by Bio-Plex Cytokine panel.

We rewrite the discussion of the manuscript, linked the results of canonical signaling pathway analysis and cytokine analysis, and also mentioned studies that support the similar results with us.

It is a good suggestion to us, we should do more confirm assay to support the postulation. There is really no direct evidence to verify the results of the signaling pathway, but we found the consistent expression trend of canonical signaling pathway and cytokine analysis. For example, the IL-17A signaling in airway cells was down regulated in H7N9-infected NHBE cells in IPA analysis, also the cytokine level of IL-17 was suppressed in the data, whereas H1N1 was unaffected.

Point 4: In discussion, the lines 319-323 was not support the statements from lines 309-318.

Response 4: We rewrite the discussion of the manuscript, mentioned more information in it.

Point 5: The discussion of cytokines findings with canonical pathway is weak or no linked.

Response 5: We rewrite the discussion of the manuscript, mentioned more information in it. For example, the IL-17A signaling in airway cells was down regulated in H7N9-infected NHBE cells in IPA analysis, also the cytokine level of IL-17 was suppressed in the data, whereas H1N1 was unaffected.

“The IL-17A signaling in airway cells was down regulated in H7N9-infected NHBE cells in IPA analysis, and IL-17 was negatively expressed in our cytokine analysis. The result is consistent with other studies, Bao et al. [23]claimed that the IL-17A protein and mRNA levels were decreased in patients with H7N9 infection and a restored Th17 and Tc17 cell frequency might serve as a biomarker for disease progression in patients infected with this virus.”

Minor:

Point 6: Please improve the introduction with using a number or range instead of “many” or “frequently”. For example, line 35 and 44.

Response 6: We added some contents in the introduction:

“The pdmH1N1 virus is transmitted efficiently among humans and there were many patients with severe pdmH1N1 infection because of the widespread transmission [2]. CDC estimated that there were about 151,700-575,400 people worldwide died from pdmH1N1 infection during the first year the virus [3].”

“Avian influenza A H7N9 virus infections frequently lead to acute respiratory distress syndrome, multiorgan disfunction, shock and death in humans, there were 133 labora-tory-confirmed H7N9 virus cases in March to July 2013, with 43 deaths reported, giving a case fatality of 32%. All fatal cases experienced pneumonia and acute respiratory distress syndrome [10]. To date, there have 1,564 confirmed cases of H7N9 infections with at least 612 deaths [11].”

Point 7: Since a subject donor has been indicated in section 2.1, please include the IRB approval.

Response 7: We mentioned the IRB approval in the text.

“Institutional Review Board Statement: The study was approved by the Institutional Review Board at the Chang Gung Medical Foundation, Taoyuan, Taiwan. (201602000A3 and 201900974B0).”

Point 8: Should not a new paragraph at line 274.

Response 8: We deleted the space at line 274.

Reviewer 2 Report

The respiratory Influenza A Viruses (IAVs) pose a significant threat to human and animal health. To accelerate our understanding of the host–pathogen response to respiratory viruses, the use of more complex in vitro systems such as normal human bronchial epithelial (NHBE) cell culture models has gained relevance as an alternative to in vivo models. In this study, authors have carried out transcriptomic analysis of host cellular responses, using NHBE cells, upon infection with 2009 pandemic IAV H1N1 (pH1N1), which is currently circulating as a seasonal virus and the avian influenza virus H7N9. In addition, differential cytokine expression after infection with pH1N1 or H7N9 viruses was also evaluated. Significantly different gene or cytokine expression patterns were found between pH1N1 and H7N9 infected NHBE cells. Data from the manuscript suggest that H7N9 virus infection induced a strong immune and likely inflammatory response, while inhibiting cellular repair mechanisms. The manuscript is a descriptive work that provide interesting data, although the novelty of the result is limited. In addition, further analysis to confirm the biological relevance of their findings were not conducted. There are some specific comments that authors should address.

  • Authors have used for their studies, NHBE cells from one single donor. This limit the impact of the study and could lead to misinterpretation of the data, assuming as the rule the observation from one single subject. Authors could consider perform additional studies using NHBE cells from other donors, at least to confirm the main findings of the study.
  • The H7N9 virus used, is a high or low pathogenic strain? Please, include this information in the manuscript.
  • Please include additional information regarding the assay to detect the virus. Eg. Which sequence is recognized by primers and probe? Is the sequence conserve in both viruses? What kind of plasmid was used as control? Are they measuring vRNA?
  • Authors should evaluate the amount of NHBE cells that are really infected, for instance by immunofluorescence or flow Cytometry. Differences in replication (Fig.1) can be due to that H7N9 infect more cells.
  • In addition, it is unclear how many times (and the number of replicates used) the assay in figure 1 was carry out. Please, include also the SD and information regarding the statistical analysis and the meaning of **/***.
  • If possible, authors should increase the size and resolution of figures 4 and 5.
  • In the point 3.4 (Differential Cytokine Expression in NHBE Cells Challenged by pdmH1N1 or H7N9), why authors have selected those short time points (4, 8, 12 hpi)? It is not better use the same time points than in the transcriptomic analysis? 4-12 hpi seems to be short times as compared with other studies in the literature.
  • Author should consider include additional information in the discussion, mentioning the main hits/genes/cytokines that have been identified. In fact, 15 references in the entire manuscript is not too much.
  • In the manuscript is indicated: Supplementary Materials: The following are available online at www.mdpi.com/xxx/s1, Figure S1: 387 title, Table S1: title, Video S1: title. However, the reviewer cannot see the sup material, and this is not mentioned in the text.

Author Response

Response to Reviewer 2 Comments

Point 1: Authors have used for their studies, NHBE cells from one single donor. This limit the impact of the study and could lead to misinterpretation of the data, assuming as the rule the observation from one single subject. Authors could consider perform additional studies using NHBE cells from other donors, at least to confirm the main findings of the study.

Response 1: In this study, we mainly wanted to compare the differences between pdmH1N1 and H7N9 under the same infection conditions, so cell lines from multiple donors were not used.

In our early study published in 2018 , A pilot study on primary cultures of human respiratory tract epithelial cells to predict patients' responses to H7N9 infection (https://www.ncbi.nlm.nih.gov/pmc/articles/PMC5865685/), we compared the H7N9 and pdmH1N1 infection in NHBE cells from 24-year-old and 69-year-old female donors, and also in primary epithelial cells (harvested from 27 patients undergoing airway surgery). We found that viral RNA quantity at 72 h was significantly higher in patients aged 21–64 years than in patients aged ≥ 65 years; however, no effects of sex, medical comorbidities, and obesity were noted. We found that a donor's age might have an effect on viral RNA quantities (H7N9 and H1N1pdm) and cytokine levels (IL-1β, IL-8, IFN-γ, IP-10, and TNF-α) of the commercial NHBE culture supernatants, and certain patient-related characteristics could modulate viral replication and the cytokine response (e.g. age ≥ 65 years could decrease viral RNA quantity and the levels of IL-1β and IL-8, and increase the IL-6 level; male sex increased the levels of IL-1β, IL-6, IL-8, and IFN-γ; medical comorbidity increased the IL-8 level; and obesity increased the IL-8 level at 24 h p.i., and decreased the IL-8 level at 72 h p.i., increased the changes in viral RNA quantity, and decreased the changes in IL-1β and IP-10 for H7N9 infection in human respiratory tract primary epithelial cells.

Whether H7N9 or pdmH1N1, the cytokine expression level at 69-year-old NHBE is slightly weaker than that in 24-year-old NHBE, but the trend is similar (up or down).

Point 2: The H7N9 virus used, is a high or low pathogenic strain? Please, include this information in the manuscript.

Response 2: We include more information about the virus we used in the manuscript. “A/Taiwan/4-CGMH2/2014 (H7N9) is genetically close to the A/Anhui/1/2013 strain, which was the fourth H7N9 infection reported on April 25, 2014 in Taiwan [12].” This case is a businessman with a history of travel to China, and recovered after hospitalization with clinical treatments. The virus is a high pathogenic strain.

Point 3: Please include additional information regarding the assay to detect the virus. Eg. Which sequence is recognized by primers and probe? Is the sequence conserve in both viruses? What kind of plasmid was used as control? Are they measuring vRNA?

Response 3: We included more information about the protocol to detect the influenza A virus. “The primers and probe that targeted the conserved sequences on matrix gene (M) of influenza A used in this study were originally developed by the World Health Organization Collaborating Centre in Beijing, China (2013).”

For quantification, we used plasmid DNAs cloned with matrix gene of influenza A virus at six different concentrations, ranging from 10–106 copies/μL, were run in parallel with all samples. And we measured the quantities about vRNA.

Point 4: Authors should evaluate the amount of NHBE cells that are really infected, for instance by immunofluorescence or flow Cytometry. Differences in replication (Fig.1) can be due to that H7N9 infect more cells.

Response 4: We used the same amounts of NHBE cells and the same titers of the two viruses and got the MOI of 3 to do the assay of figure 1, so we found that the growth of H7N9 can reach higher titers than pdmH1N1.

The inoculate virus titers are same for both viruses, we infected both viruses at a multiplicity of infection of three. We modified the statements about result 3.1 and Figure 1, we removed the “growth kinetics” in the text.

“3.1. Viral RNA quantities comparison of Influenza A pdmH1N1 and H7N9 in NHBE Cells

We compared the viral RNA quantities of influenza A pdmH1N1 and H7N9 in NHBE cells (Figure 1). NHBE cultures were infected with each virus at an MOI of 3, and culture supernatants at 4, 8, 12 and 36 h postinfection (hpi) were collected and viral RNA copies was evaluated using qRT-PCR. We found a higher viral load of pdmH1N1 than H7N9 at 4 hpi, with a p-value 0.0293. But a higher viral load of H7N9 than pdmH1N1 at 8, 12 and 36 hpi, with p-value 0.0177, 0.0075 and 0.0104 respectively. Indicating that the growth of H7N9 can reach higher titers than pdmH1N1.”

Point 5: In addition, it is unclear how many times (and the number of replicates used) the assay in figure 1 was carry out. Please, include also the SD and information regarding the statistical analysis and the meaning of **/***.

Response 5: We mentioned more details of the assay of figure 1. In the manuscript.

“Figure 1. Normal human bronchial epithelial (NHBE) cell cultures were infected with pandemic influenza H1N1 (pdmH1N1) and H7N9 at a multiplicity of infection of three and performed in triplicate. Culture supernatants at 4, 8, 12 and 36 h postinfection (hpi) were collected and the viral load was evaluated using quantitative real-time reverse transcription polymerase chain reaction. (*=p-value 0.01-0.05, Significant; **=p-value 0.001-0.01, Very significant)”

Point 6: If possible, authors should increase the size and resolution of figures 4 and 5.

Response 6: We adjusted the resolution of figure 3 and figure 4 and the size of figure 5.

Point 7: In the point 3.4 (Differential Cytokine Expression in NHBE Cells Challenged by pdmH1N1 or H7N9), why authors have selected those short time points (4, 8, 12 hpi)? It is not better use the same time points than in the transcriptomic analysis? 4-12 hpi seems to be short times as compared with other studies in the literature.

Response 7: It is a good suggestion to use the same time points than in the transcriptome analysis. We observed differential cytokine expression in NHBE cells challenged by pdmH1N1 or H7N9 on short time points, so we didn’t do more cytokine analysis on long time point.

Point 8: Author should consider include additional information in the discussion, mentioning the main hits/genes/cytokines that have been identified. In fact, 15 references in the entire manuscript is not too much.

Response 8: We rewrite the discussion of the manuscript, mentioned more information in it. And the references in the manuscript was added to more than 30.

Point 9: In the manuscript is indicated: Supplementary Materials: The following are available online at www.mdpi.com/xxx/s1, Figure S1: 387 title, Table S1: title, Video S1: title. However, the reviewer cannot see the sup material, and this is not mentioned in the text.

Response 9: We removed the “Supplementary Materials: The following are available online at www.mdpi.com/xxx/s1, Figure S1: 387 title, Table S1: title, Video S1: title.” In the manuscript.

Round 2

Reviewer 1 Report

Please check and confirm ALL the contents in table 2. There were 2 FLG in down regulation of H1N1. 

No other comment.

Author Response

Response to Reviewer 1 Comments

Point 1: Please check and confirm ALL the contents in table 2. There were 2 FLG in down regulation of H1N1. 

Response 1: We have removed one of the repeating gene FLG and checked the contents about the top 10 up and down DEGs in table 2.

Reviewer 2 Report

The manuscript is suitable for its publication. 

Author Response

Response to Reviewer 2 Comments

Point 1: The manuscript is suitable for its publication. 

Response 1: Thank you for the good comments and suggestions.